# Trends in Intake, Length of Stay and Outcome Data in a Portuguese Animal Shelter Between 2018 and 2024

**DOI:** 10.3390/ani16010141

**Published:** 2026-01-04

**Authors:** Rui Jardim, Bruno Colaço, Maria de Lurdes Pinto, Sofia Alves-Pimenta

**Affiliations:** 1School of Agrarian and Veterinary Sciences, University of Trás-os-Montes and Alto Douro (UTAD), 5000-801 Vila Real, Portugal; 2Veterinary and Animal Science Research Centre (CECAV), Associate Laboratory for Animal and Veterinary Sciences (AL4AnimalS), University of Trás-os-Montes and Alto Douro (UTAD), 5000-801 Vila Real, Portugal; 3Centre for the Research and Technology of Agro-Environmental and Biological Sciences (CITAB), Inov4Agro, University of Trás-os-Montes and Alto Douro (UTAD), 5000-801 Vila Real, Portugal

**Keywords:** animal welfare, adoption, live release, shelter management, public policies

## Abstract

Overpopulation in municipal shelters is a significant challenge across Europe. This study analyzed 2291 dog and cat admissions to a Portuguese municipal shelter between 2018 and 2024, covering the COVID-19 pandemic. For the 1325 adoptable animals, Length of Stay proved critical: adoptions were most frequent within the first week, while prolonged stays significantly reduced success rates. Although microchipped animals were 71 times more likely to be returned to owners, only 9.5% arrived with identification, revealing low compliance with mandatory law requirements. While the Live Release Rate (79.7%) and Save Rate (84.7%) reflect a strong commitment to life-saving outcomes, a Population Balance Calculation of 75.6% indicates cumulative overcrowding. Collaborations with animal protection associations were vital in finding homes for hard-to-place animals, particularly seniors and cats. These findings highlight the need for early adoption initiatives, improved microchip compliance, and species-specific strategies like Trap-Neuter-Return. This study offers a data-driven framework to guide evidence-based shelter management in Portugal and similar European systems within a One Health perspective.

## 1. Introduction

The growing challenge of managing stray, abandoned, or surrendered companion animals has placed considerable strain on municipal animal shelters across Europe [1]. Since 2018, following the implementation of Law nº 27/2016, euthanasia of healthy animals due to space constraints has been prohibited in Portuguese municipal shelters, being permitted only in cases of incurable disease, suffering, or serious risk to others [2]. While reinforcing ethical animal welfare standards, this legislation has intensified operational pressures on shelters, making evidence-based management strategies essential [3]. Rising intake, particularly of stray and free-roaming animals, creates chronic capacity challenges and prolonged stays, threatening adequate welfare care. Without robust adoption, sterilization, and public education initiatives, stronger welfare protections may inadvertently increase shelter burdens [4,5].

Companion animal ownership is widespread across Europe, with approximately 139 million households (49%) owning one or more pets, including around 90 million dogs and 108 million cats [6]. In Portugal, an estimated 2.58 million dogs and 1.96 million cats are owned nationally [7]. The COVID-19 pandemic further affected animal welfare dynamics in shelters. While municipal shelters experienced increased operational pressures, non-governmental animal associations faced more pronounced challenges, including financial constraints and volunteer shortages, underscoring the pandemic’s variable impact on shelter capacity and animal care [8,9]. Despite these challenges, Portuguese municipal shelters demonstrated resilience, maintaining more stable operations than non-governmental organizations [5,10].

Standardization of shelter key performance indicators (KPI) remains a challenge worldwide. Although the Live Release Rate (LRR) is widely used, inconsistencies in its definition and calculation limit comparability. Harmonized metrics, including Length of Stay (LOS), Save Rate (SR), and Population Balance Calculation (PBC), have been recommended to improve benchmarking, evidence-based management, and policy development [11,12].

Several countries have developed robust data infrastructures: in the United States, the Shelter Animals Count initiative provides a centralized national database, while Spain’s Fundación Affinity publishes annual adoption and abandonment trends [13,14]. The 2025 Hill’s State of Shelter Pet Adoption Report provides further insights into adoption dynamics and persistent barriers in the U.S [15].

In other countries, like Portugal, however, comprehensive standardized data remain limited [16]. A 2023 national census estimated 931,556 free-roaming animals, including 830,541 cats and 101,015 dogs [17], while 2024 municipal shelter records reported 42,718 intakes, with 30,126 adoptions and 2421 euthanasias [18]. Limited harmonized data constrain epidemiological analyses and evidence-based decision-making at the municipal level [17].

Adoption rates from shelters remain low. Surveys indicate that only a small fraction of adopted pets (16.2%) originate from shelters, with the majority adopted directly from streets or offered by friends as well as from strangers, mostly through the internet, potentially bypassing microchipping and reducing adoption-related responsibilities [19]. Geographic and socioeconomic profiles influence shelter dynamics: animals often originate from areas of higher vulnerability, whereas adopters tend to reside in more affluent communities, reflecting structural inequities in abandonment and adoption patterns [20]. Owner-related factors, such as financial constraints, housing issues, behavioral challenges, and family changes, remain key drivers of relinquishment [21,22].

Shelter outcomes are influenced not only by animal- and adopter-related factors but also by institutional practices, staffing, and facility resources [23]. Understanding these systemic influences is essential to develop effective strategies that increase adoption rates and improve overall animal welfare [22,24]. Studies have demonstrated that shelter outcomes vary significantly across different management systems and cultural contexts [25,26]. Comprehensive reviews of shelter interventions demonstrate that adoption outcomes depend on complex interactions between animal characteristics, shelter practices, and community factors [27].

This study aims to characterize the profiles, intake patterns, and outcomes of dogs and cats admitted to a Portuguese animal shelter (CVM Feira) between 2018 and 2024, describing demographic trends, adoption predictors, and the impact of LOS on shelter capacity. By providing a seven-year longitudinal analysis including the COVID-19 pandemic period, this study establishes an evidence-based framework to support municipalities in improving animal welfare outcomes and guiding public policy, while offering insights relevant to comparable European shelter systems.

## 2. Materials and Methods

### 2.1. Study Design and Population

This retrospective observational study analyzed the complete intake records of animals housed at the Santa Maria da Feira Municipal Veterinary Centre (CVM Feira) and its partner network (CIAMTSM and APA) between 1 January 2018, and 31 December 2024. Records were excluded based on the following criteria:Missing mandatory microchip information for stray animals.Dogs involved in bite incidents, that were not presented to CVM Feira despite police notification.Animals registered for external adoption that escaped from the shelter prior to intake, admission, or adoption.Animals intended for external adoption that died before intake.

After applying these exclusion criteria, the final dataset was used for further analysis. As a local municipal shelter, CVM Feira does not receive animals transferred from other shelters. However, due to ongoing intake demands and limited housing capacity, it regularly transfers animals to other facilities. The shelter works closely with CIAMTSM and collaborates with non-profit animal protection associations (APA) to enable centralized record collection and to expand territorial coverage.

All animals were assigned to intake categories adapted from the Shelter Animals Count (SAC) framework [12] and guided by recent studies on shelter populations [28,29]. Intake categories were harmonized with international definitions but adjusted to reflect CVM Feira’s operational realities. While SAC often treats processes such as Return-to-Owner (RTO) or Return-to-Field (RTF) primarily as community services, at CVM Feira these animals had formal entry and exit records and were therefore included in the dataset as both intake and outcome events. It is acknowledged that including RTF animals in intake statistics may influence shelter performance metrics, as these animals are not available for adoption and have predetermined outcomes. Similarly, animals initially found as strays by local citizens or APA and temporarily kept under their care, or those registered for external adoption, were formally surrendered for veterinary procedures and subsequently adopted. These cases were classified as Owner or APA surrender at intake and as adoption at outcome.

Each intake event was recorded independently under a unique identifier. The internal database was not designed to systematically flag repeated admissions of the same animal (e.g., following unsuccessful adoptions or returns), and therefore recidivism could not be reliably identified or quantified in this analysis.

#### Shelter Facilities and Operational Practices

The CVM Feira functions as an Official Municipal Animal Shelter (CRO) and operates in close collaboration with the Official Intermunicipal Animal Shelter of the Association of Municipalities of Terras de Santa Maria (CIAMTSM) and several non-profit animal protection associations (APA). CVM Feira has 12 dog kennels and a small core team (one veterinarian, one caretaker and one administrative staff member), which limits on-site housing capacity. Animals requiring longer stays or additional space are transferred to CIAMTSM, whose main facility comprises 51 dog kennels distributed across four wings, three semi-circular quarantine kennels, an additional 10-kennel wing, and 2 dog pack parks for group housing of dogs, which became operational in 2021.

Newly admitted dogs are placed in clean, disinfected kennels, and animals are periodically relocated between kennels to allow thorough cleaning and disinfection before receiving new intakes. Behavioral assessment is based on veterinarian and caretaker observations during routine handling and is used to inform housing and adoption counselling. According to local protocols, animals adopted from the municipal system are routinely dewormed internally and externally, vaccinated according to age, identified with a microchip (if not already identified), and sterilized (spayed or castrated) when clinically appropriate, either before adoption or shortly thereafter.

### 2.2. Data Collection and Management

Data were initially collected and organized using the internal shelter database *CVM Animais Feira.xlsx* (Google Sheets; Google LLC, Mountain View, CA, USA). A refined and structured dataset was generated for analysis (CVM Feira_2018_2024_db.xlsx). The national Companion Animal Information System (SIAC, www.siac.vet) was used to cross-verify microchip data, ownership history, and final outcome dates. Additional information on animals transferred to the Intermunicipal Animal Shelter of Santa Maria da Feira (CIAMTSM) was obtained from the PetCare shelter management platform (v1.16.2; Wiremaze, Matosinhos, Portugal, https://www.wiremaze.com/o-que-fazemos/equipas/petcare (accessed on 12 October 2025), which is used by that facility. Data on transfers to associations were obtained exclusively from the national companion animal database (SIAC) and from microchip transponder queries, including attached ownership transfer documents and adoption dates where available.

All intake and outcome records were standardized to ensure consistent tracking of individual animals throughout their shelter trajectory and a transparent overview of data compilation. Each animal was assigned a unique identifier number, and outcomes were classified into mutually exclusive categories, enabling accurate calculation of adoption rates, length of stay, and live release metrics [29].

The study relied exclusively on anonymized administrative data and involved no interventions with animals.

### 2.3. Variable Definitions and Inclusion Criteria

#### 2.3.1. Intake Variables

Demographic: Species (dog/cat); Sex (male/female); Age at intake (puppy/kitten: ≤6 months; young adult: >6 months to ≤2 years; adult: >2 years to ≤8 years; senior: >8 years) [28]; Breed group (mixed-breed, recognized breed, or potentially dangerous breed, under Portuguese law [30]; and Body size (dogs only: small, medium, large) [31].○Operational definition of breed group: Breed group classification was applied to dogs and cats using a three-category approach: mixed-breed, recognized breed, and potentially dangerous breed. Under Portuguese law (Decree-Law No. 276/2001, consolidated version), “purebred” animals are those identified and registered with a genealogical record in the national studbook (Livro de Origens Português) [32]. In this dataset, none of the animals had studbook registration. For the purposes of this study, all animals without objective pedigree documentation or with uncertain breed identity were therefore grouped in the mixed-breed category, which corresponds to legally undefined-breed animals and represents the majority of the sample. Animals were classified as a recognized breed when their phenotype and available documentation were consistent with a specific breed, based on the attending veterinarian’s visual assessment at intake and, when available, information provided by the owner or caretaker and breed data recorded in the national SIAC database for microchipped animals. International breed standards were used as conceptual references for this classification, in particular Fédération Cynologique Internationale (FCI) standards for dogs and Fédération Internationale Féline (FIFe) standards for cats, as applied in national kennel and feline registries. Breed information was recorded and communicated to potential adopters only at this aggregated level (mixed-breed, recognized breed, potentially dangerous breed), and no breed-by-breed analyses were conducted.○Operational definition of body size: Body size classification was applied to dogs only, which were assigned to three size groups, following the height-at-the-withers categories adapted from FCI agility regulations: small (<35 cm), medium (35–<48 cm), and large (≥48 cm). The “medium” category in this study therefore aggregates the FCI medium (35–<43 cm) and intermediate (43–<48 cm) classes into a single group for analytical purposes [31].Administrative: Year of Intake; Microchip presence at intake; Total Intake (number of animals entering the shelter during a year) and Intake Type Categories [33]:○Stray: Animals collected from public spaces by local citizens or municipal services, as well as free-roaming cats admitted through the Trap–Neuter–Return (TNR) program [34].○Owner or APA surrenders: Voluntary relinquishment by owners, accredited animal protection association (APA) surrenders, and animals initially rescued or managed by local citizens, municipal services, or APA and later reported to the shelter. When verified as free roaming, these externally rescued animals were accepted by CVM Feira and underwent the same care, treatment, and adoption procedures as animals physically admitted to the shelter. This category also included animals voluntarily delivered by owners or taken under police order following bite incidents involving humans or other animals, in accordance with the 15-day legal holding period established by Ministerial Order n. º 264/2013 [35].○Seized: Animals taken into custody following legal or official intervention (e.g., in cases of suspected animal abuse), corresponding to the SAC *Seizure* intake. The term *Seized* is used here as in to avoid ambiguity with the medical term *Seizures* [28].

#### 2.3.2. Outcome Variables and Performance Metrics

Adoption Status: A binary variable (adopted vs. not adopted). The “not adopted” category included animals with other final outcomes [29].Other Outcome Categories:○Return-to-Owner (RTO): Animals returned to their previous owners.○Return-to-Field (RTF): Free-roaming animals returned to their original location after being sterilized at the CIAMTSM, typically as part of TNR programs.○Died: Animals that died in the shelter, either from natural causes or by euthanasia.○Still housed: Animals remaining in the shelter at the end of the study period.Live Outcomes: Included adoptions, RTO and RTF. Transfers out to partner shelters (CIAMTSM and APA) were tracked throughout the animal’s trajectory. Only final outcomes (adoption, RTO, RTF, died, or still housed) at the last shelter where the animal resided were included in outcome calculations to avoid double counting.Length of Stay (LOS): Defined as the number of days between intake and outcome, analyzed both as a continuous and categorical variable [36]. LOS categories were based on legal and operational timeframes:○Fast Track: ≤7 days.○>Normal Track: 8–15 days.○Slow Tracks:▪I: 16–60 days.▪II: 61–180 days.▪III: 181–365 days.○Chronic Tracks:▪I: 366–1095 days.▪II: >1095 days.

The 7-day Fast Track threshold aligns with optimal adoption flow recommendations in shelter medicine, while the 15-day Normal Track corresponds to the legal period during which found animals may be claimed by owners in Portugal [37]. Subsequent categories reflect progressive stages of chronicity, which are associated with declining adoption prospects and increasing welfare concerns.

Shelter Key Performance Indicators (KPI)—Performance Metrics:○Live Release Rate (LRR): LRR was calculated following the outcome-based definition used by Shelter Animals Count (SAC), as the proportion of live outcomes (adoptions, RTO and RTF) divided by total outcomes (live and non-live outcomes), excluding animals still housed at the end of the reporting period. This outcome-based approach is widely adopted in the sheltering community and facilitates comparison with existing benchmarks [12]. An alternative, intake-based formulation defines LRR as the proportion of animals leaving alive divided by total intake (as used, for example, in the ASPCA LRR), which places greater emphasis on population flow and the contribution of new admissions to live outcomes. A high LRR is widely recognized as a primary indicator of a shelter’s commitment to saving lives and is a common benchmark in the animal welfare community [29].○Save Rate (SR): SR was calculated as the total number of live outcomes divided by total intake, providing an intake-based measure of the proportion of animals entering the shelter that did not experience a non-live outcome [12]. It should be noted that while SR reflects the proportion of animals that avoided non-live outcomes, it does not indicate what proportion of animals had an outcome of any kind, thereby not accounting for animals who remain in long-term shelter care.○Population Balance Calculation (PBC): Calculated as the total number of animals leaving the shelter (live or non-live outcomes) divided by the Total Intake. A PBC of 100% indicates population stability, while values below 100% suggest population growth, and values above 100% indicate a net population decrease [12].

### 2.4. Statistical Analysis

All statistical analyses were performed using IBM SPSS Statistics v24. Descriptive statistics were calculated for quantitative variables. The normality of data was assessed using the Shapiro–Wilk test and histogram inspection. When a non-normal distribution was confirmed, the median and interquartile range (IQR) were used instead of the mean and standard deviation.

Pearson’s Chi-Square tests were applied to compare categorical variables. If >20% of cells had expected values below 5, the Fisher-Freeman-Halton Exact test was used. Post hoc analyses were conducted using standardized residuals [38].

To explore trends in the shelter population across the five-year study period, linear regression analyses were performed for intake and outcome data (Total Intake, Total Live Outcomes, Still Housed, Died) and for KPIs (LLR, SR, and PBC) [29]. Prior to conducting the analyses, the assumptions of linear regression were tested for our count and percentage/ratio data by examining normal Q–Q plots. It was determined that the data met the assumptions of linear regression.

Binary logistic regression models were used to describe the likelihood of adoption for adoptable animals (*n* = 1325) based on age at intake group, LOS categories, intake type, presence of microchip at intake, sex, body size and breed group. Adoptable animals were defined as those either adopted during the study period (*n* = 780) or still housed in the shelter as of 31 December 2024 (*n* = 545). Animals with outcomes that rendered them unavailable for adoption, such as return to owner (RTO), return to field (RTF), died, or escaped, were excluded from this analysis. Statistical significance was set at *p* < 0.05 without adjustment for multiple comparisons [39,40].

## 3. Results

### 3.1. Demographic Characteristics and Intake Patterns

Between 2018 and 2024, 2291 animals were registered at the CVM Feira. Fifty-three records (2.3%) were excluded based on the criteria described in the Section 2: missing microchip data (*n* = 26), escapes or missing before intake (*n* = 12), external adoptions involving animals that escaped (*n* = 8) or died (*n* = 4) prior to admission, and owner surrenders related to bite incidents not presented to the CVM Feira despite police notification (*n* = 3). The final dataset comprised 2238 animals, 1110 dogs (49.6%) and 1128 cats (50.4%), used for subsequent analyses (Table 1). Intake distribution was nearly equal between dogs and cats. Puppies/kittens accounted for 42.0% (*n* = 940), adults for 32.3% (*n* = 720), young adults for 19.7% (*n* = 440), and seniors for 6.2% (*n* = 138). Females slightly predominated (51.8%, *n* = 1159). Most animals were mixed-breed (92.9%, *n* = 2078), with recognized breeds representing 6.4% (*n* = 144) and potentially dangerous breeds 0.7% (*n* = 16, dogs only).

The majority entered as strays (50.8%, *n* = 1136), followed by owner/APA surrenders (42.3%, *n* = 947) and seizures (6.9%, *n* = 155). Only 9.5% (*n* = 212; 18.6% of dogs and 0.4% of cats) were microchipped at intake. A strong association was observed between microchip presence and return-to-owner outcomes, with identified animals being 71 times more likely to be reclaimed (OR = 71.36; 95% CI: 45.05–113.04).

Annual intake volume increased substantially over the study period, rising from 119 animals in 2018 to 566 in 2024, corresponding to a 376% increase. In 2018–2019, nearly all admitted animals were dogs, but from 2020 onwards the proportion of cats increased steadily, reaching 56.9% of total entries by 2024.

### 3.2. Trends in Intake and Outcomes

The trends in overall intake and outcomes for all animals, dogs, and cats, are illustrated in Figure 1, highlighting the dynamic nature of this shelter population (*n* = 2238; Dogs: *n* = 1110; Cats: *n* = 1128) over the seven-year period. Annual intake increased significantly over time (*p* < 0.001, R^2^ = 0.602). The distribution of outcomes (live outcomes, died, still housed) differed significantly among years. Early in the COVID-19 pandemic, dog adoptions increased in the CMV Feira, and abandonment decreased, yet these trends reversed in the following year.

During the period in study, at the CVM Feira, 167 animals died of natural or unavoidable causes, and 42 were humanely euthanized due to incurable disease, unrelievable suffering, or severe aggression that compromised animal or public safety. Mortality causes were not assessed for animals housed at CIAMTSM or APA; however, all the facilities operate under a no-kill policy, and euthanasia was performed solely as a last resort on strictly medical or welfare grounds [2].

As of 31 December 2024, a total of 545 animals (313 dogs and 232 cats) remained housed across CVM Feira, CIAMTSM, and APA partner shelters. This total represents the cumulative number of animals admitted between 2018 and 2024 that had not reached a final outcome, corresponding to the sum of the ‘Still housed’ cohorts shown by intake year in Figure 1.

The overall flow of animal intake and outcomes during the study period is summarized in Figure 2 and Figure 3. Animals may be transferred multiple times before reaching their final outcome, and adoptions can occur directly from the CVM Feira, from the CIAMTSM, or from partner APA shelters. For instance, 48 dogs that were initially transferred from the CVM Feira to the CIAMTSM were, as of 31 December 2024, recorded under the responsibility of APA shelters, indicating that they had been further transferred and remained housed there (*n* = 44) or died (*n* = 4). Therefore, only the final outcomes recorded at the last shelter where the animal resided were included in this analysis to avoid double counting.

### 3.3. Predictors of Adoption

Results from the binary logistic regression analysis focused on a subset of 1325 animals considered adoptable (780 adopted; 545 still housed) are presented in Table 2.

#### 3.3.1. Sex

Sex was a significant predictor for dogs, with female dogs having higher odds of adoption (OR = 0.664; 95% CI: 0.497–0.887; *p* = 0.006) when compared to males. For cats, sex was not a significant predictor.

#### 3.3.2. Age at Intake

Initial age at intake also significantly influenced the adoption probability. Compared to puppies/kittens (≤6 months), adult animals (2–8 years) had significantly lower odds of adoption (OR = 0.617; 95% CI: 0.471–0.808; *p* < 0.001). This pattern was consistent for dogs, where adult dogs also had significantly lower odds of adoption (OR = 0.650; *p* = 0.011). For cats, adult cats similarly showed lower odds of adoption (OR = 0.463; *p* = 0.004). Figure 4 illustrates the age at intake for adopted versus still-housed animals.

#### 3.3.3. Intake Type

For all adoptable animals, those surrendered by owners or APA had significantly higher odds of adoption compared to seized animals (OR = 1.968; 95% CI: 1.350–2.868; *p* < 0.001). This effect was even more pronounced for dogs (OR = 2.233; *p* < 0.001). Figure 5 shows the intake type distribution for adopted and still-housed animals.

#### 3.3.4. Length of Stay as Primary Predictor

The LOS data was non-normally distributed, and due to its skewness, including animals with exceptionally long stays, non-parametric statistics such as median and IQR were used for descriptive analysis [41,42]. Outliers were retained, as they represent real cases of prolonged sheltering and are essential for understanding long-term population dynamics. The median LOS for all animals admitted to CVM Feira between 2018 and 2024 was 84 days (IQR: 240 days). Dogs had a considerably longer median LOS (118 days; IQR: 452 days) compared to cats (12 days; IQR: 176 days).

Animals still housed at the end of the study period (31 December 2024) were included in the LOS analysis, with their LOS calculated as the number of days from intake to the study end date. This approach provides a realistic representation of current shelter population dynamics, though it may slightly overestimate LOS for animals ultimately adopted shortly after the study period.

Within this context, LOS emerged as the strongest predictor of adoption in the adoptable population (*n* = 1325) (OR = 0.997; 95% CI: 0.996–0.997; *p* < 0.001). Compared to animals adopted within 7 days (Fast Track), the odds of adoption decreased significantly with each subsequent LOS category. Animals in Normal Track (8–15 days) had significantly lower odds of adoption (OR = 0.233; *p* = 0.028). Animals in Slow Track I (16–60 days) showed OR = 0.300 (*p* = 0.037), Slow Track II (61–180 days) showed OR = 0.116 (*p* < 0.001), Slow Track III (181–365 days) showed OR = 0.114 (*p* < 0.001), Chronic Track I (366–1095 days) showed OR = 0.035 (*p* < 0.001), and animals in Chronic Track II (>1095 days) had drastically reduced odds (OR = 0.002; *p* < 0.001).

Figure 6 illustrates how adopted animals are concentrated in shorter LOS tracks, whereas still-housed animals accumulate in longer LOS categories.

Age at intake strongly influenced LOS patterns. Puppies/kittens remained the longest (median 149 days; IQR: 209), whereas young adults had the shortest LOS (0 days), largely due to same-day external adoptions or TNR procedures. These cases were intentionally not adjusted, as they reflect the actual time spent in the shelter. Recognized breeds were adopted faster (median 19 days) than mixed-breeds (91 days). Potentially dangerous breeds also showed relatively short stays (28 days). Body size showed similar trends: large animals had the shortest LOS (24 days), followed by small (85 days) and medium-sized animals (116 days).

Intake type further shaped LOS outcomes. Stray animals typically had very short stays (3 days), whereas owner/APA surrenders (154 days) and seized animals (663 days) remained substantially longer.

Outcomes were closely associated with LOS. Adopted animals (*n* = 780) had much shorter stays (median 181 days; IQR = 458) compared with animals still housed at the end of the study (*n* = 545; median 593 days; IQR = 965). RTO animals left the shelter quickly (median 14 days; IQR = 18) and RTF animals had minimal LOS, consistent with operational procedures. A substantial proportion of still-housed dogs in December 2024 remained in Chronic Track II (>1095 days), indicating the accumulation of long-term residents.

### 3.4. Shelter Key Performance Indicators

Three KPI were calculated according to the standardized definitions provided by the Shelter Animals Count (SAC) Glossary [12]. The overall Live Release Rate (LRR) as a function of the outcomes was 79.7%, with species-specific rates of 73.3% for dogs and 85.5% for cats. The overall Save Rate (SR) was 84.7%, with cats showing a higher SR (88.5%) compared to dogs (80.8%). In contrast, the Population Balance Calculation (PBC) was 75.6%. When analyzed by species, the PBC was 71.8% for dogs and 79.4% for cats.

Figure 7 shows the trends in LRR, PBC, and SR for the combined population of dogs and cats admitted to CVM Feira from 2018 to 2024. The lower Live Release Rate (LRR) and Save Rate (SR) observed in 2018 and 2019 reflect the early phase of the shelter’s operation, when infrastructure, clinical protocols, and veterinary resources were still being established. During this period, limitations in preventive care and emergency support, combined with occasional disease outbreaks, contributed to higher mortality, primarily due to natural causes and pre-existing conditions at intake. As operational capacity improved, particularly with the implementation of standardized protocols and the relocation to the new facility in 2023, health outcomes stabilized, with LRR and SR consistently exceeding 80% in subsequent years, in line with no-kill benchmarks.

## 4. Discussion

This study presents, to the authors’ knowledge, the first detailed analysis of companion animal shelter dynamics at a municipal level in Portugal, offering critical insights into factors influencing animal outcomes and underscoring the need for data-driven management strategies. Although Portugal currently lacks a centralized national system comparable to Shelter Animals Count in the USA [13], this study demonstrates the feasibility of compiling detailed municipal-level shelter data to support evidence-based management.

### 4.1. Ethical Success and Population Management Challenges in a “No-Kill” Framework

The central finding of this study is the fundamental paradox facing Portuguese shelter management. On one hand, the system demonstrates strong ethical success, aligned with Law 27/2016, achieving a Live Release Rate (LRR) of 79.7% and a Save Rate (SR) of 84.7%. These figures, consistent with international benchmarks [29,43], reflect a clear commitment to saving lives. However, this success is weakened by a Population Balance Calculation (PBC) of 75.6%, which reveals systemic failure in animal flow-through. A PBC below 100% indicates cumulative population growth [12], meaning the shelter is unable to keep pace with the 376% increase in admissions and is effectively accumulating animals year after year. This suggests that while ‘no-kill’ legislation [2,3] is ethically commendable, it has inadvertently created a capacity crisis, transforming the shelter from a transit facility into long-term housing. Such “warehousing” raises serious welfare concerns regarding the quality of life for animals in chronic confinement [44].

The lower LRR and SR observed in 2018 and 2019 reflect the operational challenges of the shelter’s early years, including limited access to veterinary care, undefined vaccination and deworming protocols, and outbreaks of infectious diseases such as parvovirus [45]. The intake of animals in critical condition (injured, sick, or severely neglected) may have contributed to increased mortality rates during this period, though formal analysis of mortality causes was not conducted. These factors underscore the importance of standardized intake procedures and preventive healthcare to improve shelter outcomes [46].

Over time, the improvements in LRR and SR demonstrate the shelter’s growing capacity and commitment to animal welfare, supported by infrastructure upgrades, expanded housing, and comprehensive veterinary care [43,44]. Nonetheless, the increasing number of animals still housed at the end of 2024 raises concerns about the system’s long-term sustainability [29,47]. Although extended length of stay (LOS) may indicate improved survival, it also highlights the ethical dilemma of prolonged confinement and reinforces the need for strategies that reduce LOS and enhance animal flow-through [4,43,48].

In this study, we adopted the SAC outcome-based LRR to ensure consistency with international reporting standards, but we interpret it together with intake-based indicators such as SR and PBC. Different formulations of LRR capture distinct but complementary aspects of shelter functioning. Outcome-based LRR [12], as used here, focuses on the fate of animals with completed outcomes and is useful for benchmarking life-saving performance, whereas intake-based metrics such as SR and PBC place greater emphasis on population flow and can highlight the accumulation of long-stay animals (‘warehousing’) that may not be apparent when only outcome-based percentages are considered. Interpreting these indicators together is therefore essential when evaluating both ethical commitments to saving lives and the welfare implications of prolonged confinement [11].

### 4.2. Interpretation of Predictors of Adoption

**Length of Stay (LOS)**: Our regression model identifies the driver of this operational challenge: Length of Stay (LOS) is the strongest predictor of adoption. The comparison with international benchmarks is alarming. While Dutch and Czech shelters report median LOS of 26–53 days [25,49], the adopted animals in this study had a median LOS of 143 days, and those still housed had a staggering median LOS of 593 days. The model (Table 2) shows that the odds of adoption decrease by 77% (OR = 0.233) after just 8–15 days (Normal Track). This implies the shelter has a ‘first-week’ window for success. After this window, animals enter a vicious cycle: the stay in the shelter itself, which diminishes their adoptability. It is also possible that the causality operates in reverse, or bidirectionally: animals with inherently lower adoptability (e.g., due to age, health, or behavioral issues) may experience longer LOS from the outset, rather than LOS itself reducing adoptability. Clarifying these pathways would require longitudinal behavioral and health assessments, which were beyond the scope of this study [50]. Given that chronic LOS is the primary challenge, the other predictors identified in the model help define the population most at risk of “being left behind” [48].**Age at intake**: The bias against adult animals (dogs OR = 0.650, cats OR = 0.463) is a key finding, consistent with existing literature [26,51,52] and with age categories commonly used in shelter studies [28]. Despite their higher odds of adoption, puppies and kittens had the longest median LOS (149 days; IQR 209 days), which likely reflects operational factors rather than lack of adopter interest [48]. The need to complete age- and weight-dependent pre-adoption procedures (vaccination, deworming, microchipping, and sterilization), together with age-appropriate vaccination schedules and postponed sterilization until animals reach a suitable weight or age, can delay adoption readiness and prolong shelter stays for juveniles [45]. This juvenile category also covers a broad intake age range (0–6 months), so some animals may enter as neonates requiring intensive care or foster placement, while others arrive closer to typical adoption age, which should be kept in mind when interpreting the mean LOS of 149 days for this group. Adopter preferences within this band may also contribute, as very young puppies and kittens are often favored over slightly older juveniles, in line with previous work showing a general preference for younger shelter animals [53,54]. In contrast, many young adults typically enter already meeting core adoption requirements (or can be prepared more rapidly), helping to explain their shorter LOS, whereas older animals, particularly senior cats, continue to face adoption barriers linked to health and longevity concerns [48,51].**Intake Type**: Owner/APA surrendered animals had double the odds of adoption compared to seized animals. Seized animals had the longest median LOS of all (663 days). This reflects their dual burden: physical/psychological rehabilitation [55] and legal impediments that keep them as “living evidence” [50,56].**Breed Group**: Animals of recognized breeds had a shorter median LOS (19 days; IQR: 195 days) compared to mixed-breed animals (91 days; IQR: 242 days), which is consistent with studies suggesting adopter preferences for specific, recognizable breeds and for animals perceived as having more predictable adult characteristics, such as size, temperament, and care requirements [22]. In this shelter system, breed information was communicated to adopters only at an aggregated level (mixed-breed vs. recognized breed vs. potentially dangerous breed), but even this level of classification may shape adopter expectations [54]. Dogs classified as potentially dangerous breeds under Portuguese legislation also showed relatively short LOS among adoption-track animals (median 28 days) [30,57]. This finding diverges from reports in other countries, where restricted breeds are often more difficult to rehome and have longer LOS. In our context, it should be interpreted in light of the small number of dogs in this category, which limits statistical robustness, and of targeted efforts by the municipal shelter network and APA partners to promote these dogs through focused communication, transfers, and adoption counselling. Local adopter preferences and housing conditions may further modulate demand for specific breed types, making direct comparison with other national settings difficult.**Sex**: The preference for female dogs (with males having an OR = 0.664) reflects public preferences that may be shaped by widespread sterilization practices and cultural perceptions of manageability [20]. This pattern aligns with most jurisdictions where elective neutering is permitted, with the notable exception of Norway, where routine spaying is restricted by law [58].

### 4.3. Species-Specific Intake Trends and Management Implications

The transition from predominantly dog intakes in 2018–2019 to an increasing proportion of cats from 2020 onwards likely reflects both national and local developments. National implementation and expansion of trap–neuter–return (TNR) programmes and the formal registration of community cat colonies increased the visibility of free-roaming cats within official records [37]. Locally, partnerships with the CIAMTSM and APA shelters have facilitated the intake and placement of cats and senior animals, partially compensating for the absence of a dedicated municipal cattery. These factors together help explain the growing representation of cats in the municipal shelter system over time.

The higher LRR for cats (85.5%) compared to dogs (73.3%) is largely attributable to the Trap-Neuter-Return/Return-to-Field (TNR/RTF) program, which constituted a substantial proportion of cat live outcomes. When community cats returned to field are excluded and only adoption-track animals are considered, species-specific adoption dynamics still differ between dogs and cats, underscoring the need to interpret aggregate LRR with caution [59]. This distinction is important when benchmarking shelter performance, as TNR/RTF programs represent a fundamentally different intervention pathway from traditional adoption, and outcome statistics for community cats and adoption-track populations should be reported separately whenever possible [34,48,60].

It is also important to consider how local intake classification practices influence the interpretation of these findings. In this municipal–intermunicipal system, animals initially found as strays by citizens or APA and temporarily kept under their care, or those registered for external adoption, were formally surrendered to CVM Feira for veterinary procedures and subsequently adopted and were therefore coded as Owner or APA surrenders at intake. This approach may affect comparability with shelters using stricter SAC definitions, as it can underestimate the proportion of true stray intakes and overestimate owner-initiated surrenders, even though it accurately reflects the operational reality of animals formally transferred through community partnerships [20].

### 4.4. Microchipping Policy: Implementation Challenges and Implications

The study reveals that the overcrowding crisis is exacerbated by a critical failure at the intake front. Return-to-Owner (RTO) was the most efficient outcome pathway, and microchipped animals were 71 times more likely to be returned to their owners, in line with previous findings [33,61]. However, the prevalence of microchips at intake was just 9.5%, despite mandatory national legislation [62]. This result highlights persistent challenges in ensuring effective implementation of the microchipping policy, which may include gaps in public awareness, affordability, and enforcement. Strengthening these aspects could significantly increase RTO rates, reduce shelter overcrowding, and improve animal welfare outcomes [63,64].

### 4.5. Interpretation of Shelter Key Performance Indicators

Our results support the importance of adopting standardized shelter KPI to allow for consistent and meaningful comparisons across facilities [29,43]. Relying solely on the Live Release Rate (LRR) provides an incomplete and potentially misleading picture of shelter performance, often obscuring structural issues such as overcrowding. The adoption of standardized KPI reporting, including LRR, SR, and PBC, is therefore essential [12]. Only by transparently measuring population flow (PBC), and not just live outcomes (LRR), can municipalities make evidence-based management and funding decisions to ensure a truly humane and sustainable animal welfare system.

### 4.6. COVID-19 Pandemic and Socioeconomic Challenges

External factors, including financial constraints, housing insecurity, and limited access to veterinary care, play a significant role in adoption outcomes. Those challenges were exacerbated during the COVID-19 pandemic, between 2020 and 2021 [9,65]. Early in the pandemic situation, we observed an increment in dog adoptions in the CMV Feira, and a decrease in the abandonment decreased, yet these trends reversed in the following year, highlighting the complex, temporal effects of COVID-19 also referred to by other authors on companion animal welfare [8,66]. COVID-19-related disruptions may have affected intake and outcomes, though municipal shelters showed relative resilience compared to private associations [5,10]. Vulnerable communities often experience higher intake rates due to limited pet retention resources and reduced adoption capacity as documented in other countries [20,24]. These socioeconomic constraints including housing restrictions, financial hardship, are recognized as major drivers of pet relinquishment [22,61,67].

### 4.7. Implications for Shelter Management and Public Policy

The analyzed shelter system relies heavily on collaboration with animal protection associations (APA). The flowchart models (Figure 2 and Figure 3) show that APA are the final destination for the vast majority of still housed animals (484 of 545, or 88.8%). While this collaboration is vital, it also masks the true scale of the overcrowding. The municipal system is transferring its burden of chronic-LOS and hard-to-place animals to the non-profit sector [43]. Although the municipality provides monthly financial support to key partners via protocol, this is not a sustainable solution. It remains a transfer of the problem to partners who, despite this aid, are left with insufficient resources to manage the full burden. Future collaboration must focus not just on animal transfers but on more robust financial and logistical support for these APA to facilitate final outcomes [4].

This study’s evidence also points to clear implications for practice and policy. Given that LOS is the strongest predictor of outcomes, shelters must adopt active, risk-based flow management, using ‘fast track’ protocols and immediate interventions for at-risk animals to break the cycle of in-shelter deterioration [45,46]. Furthermore, the 71× effectiveness of microchipping for RTO highlights that enforcing this mandatory legislation as a national public health priority is essential to reduce intake [62]. Microchipping may therefore function both as a technical tool and a proxy for proactive, responsible ownership, further enhancing the likelihood of RTO for this subgroup.

Lastly, to foster a unified, data-driven approach across the country, it is strongly recommended that the national veterinary authority promotes the wider adoption of standardized KPI, including, including PBC, across all shelter types, from municipal to APA shelters. Such standardization would provide a more holistic, ‘360-degree’ view of the national situation, which is essential given that LRR alone is a vanity metric; only by transparently measuring population flow (PBC) can managers make the evidence-based decisions required for a truly humane and sustainable system [12].

### 4.8. Strengths and Limitations

Key strengths of this study include the use of comprehensive administrative records, stratified analyses by species, age, intake type, and LOS, added to the application of shelter KPI (LRR, SR, and PBC). The direct use of shelter data ensured accurate tracking of each animal’s trajectory and highlighted the preventive value of microchipping, while the combined use of bivariate and multivariate models provided robust insights into shelter dynamics and predictors of adoption.

Several limitations must be acknowledged. Behavioral and clinical data, as well as information on age at sterilization, were not consistently recorded, preventing their inclusion in models, though they likely influence adoptability [68,69].

Age at intake was categorized using a 0–6 month juvenile band [28], to facilitate comparability with previous shelter research; however, not subdividing this group into narrower age ranges (e.g., 0–3 vs. 3–6 months) may mask important differences in care needs, adopter preferences and adoption readiness within the juvenile category and should be considered a methodological limitation. Likewise, breed group classification followed a pragmatic, three-category approach based on phenotypic assessment rather than objective pedigree or genetic methods. together with context-specific management strategies and adoption promotion efforts, may have influenced the observed associations between body size, breed group and LOS and should be considered when interpreting the study findings. Sterilization status at intake was not consistently recorded in the database, limiting our ability to assess its influence on outcomes and adoption dynamics, particularly in the context of TNR program evaluation. LOS may also be slightly overestimated for some animals due to incomplete traceability in partner shelters and potential delays in updating ownership information in the national pet registration system (SIAC) [16].

The analysis focuses on a single municipal/intermunicipal system, though comparative European studies suggest similar patterns across different shelter management approaches [25,49], supporting broader applicability of these findings. However, such findings may not generalize to all regions. Small subgroup sizes, such as seniors, may reduce statistical power, and the apparent increase in still housed animals at the end of 2024 may reflect a cohort effect. Data on repeated admissions (recidivism) were not systematically captured in the database, preventing formal assessment of readmission rates. Although the structure of the municipal–intermunicipal system suggests that most relinquishments, if they occur, would be registered at downstream CIAMTSM and APA partner shelters rather than at CVM Feira, the absence of structured recidivism data remains a methodological limitation and an area for future investigation.

Future research should incorporate standardized medical and behavioral assessments, post-adoption follow-up, and cost-effectiveness analyses. Building on international benchmarks such as Shelter Animals Count and Fundación Affinity reports [13,14], developing a centralized, standardized national database for shelter intake, outcomes, and adoption trends would strengthen Portugal’s municipal shelter management and provide a reference framework for other countries and international shelter systems.

## 5. Conclusions

This seven-year analysis of a Portuguese municipal shelter system confirms that ‘no-kill’ legislation has met its ethical objective of increasing live outcomes (high LRR/SR). However, it also reveals a challenge of operational sustainability, evidenced by a low PBC and a chronically high Length of Stay (LOS), which compromises long-term animal welfare. The system is saving animals from euthanasia but condemning many to a life in confinement. The effects of the COVID-19 pandemic period may have impacted our results, increasing operational pressures.

These findings demonstrate that sustainable shelter management in Portugal, and in comparable European contexts, requires a paradigm shift from survival to flow-through. Achieving this balance demands a comprehensive, data-driven approach that integrates population management, public education, and preventive strategies such as effective microchipping enforcement and accessible sterilization programs. Strengthening collaboration between municipal authorities, veterinary services, and animal protection associations is essential to ensure consistent standards of care and transparency. At a national level, adopting standardized key performance indicators (such as LRR, SR, and PBC) would allow the creation of a unified, evidence-based framework to guide policy and funding decisions.

Ultimately, the long-term success of the “no-kill” model will depend not only on saving lives, but on ensuring quality of life, transforming shelters from places of indefinite confinement into true pathways to adoption, recovery, and responsible guardianship.

## Figures and Tables

**Figure 1 animals-16-00141-f001:**
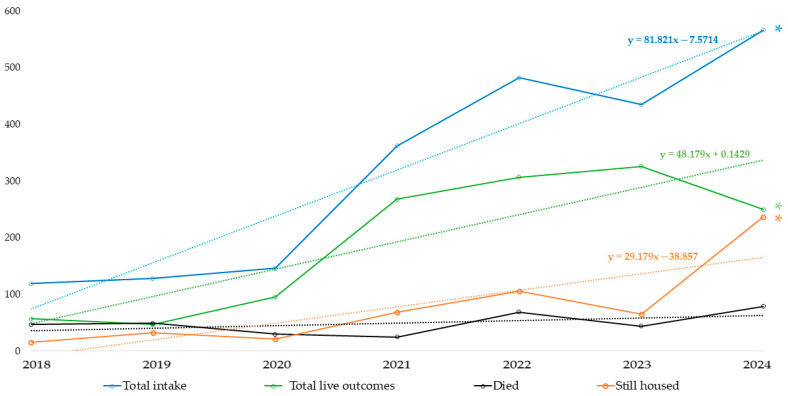
Annual trends in animal intakes and outcomes from 2018 to 2024, based on year of intake. The still housed category includes animals admitted in each year that remained in the shelter as of 31 October 2024 (*n* = 2238). * Denotes statistical significance (*p* < 0.05) in the regression analysis.

**Figure 2 animals-16-00141-f002:**
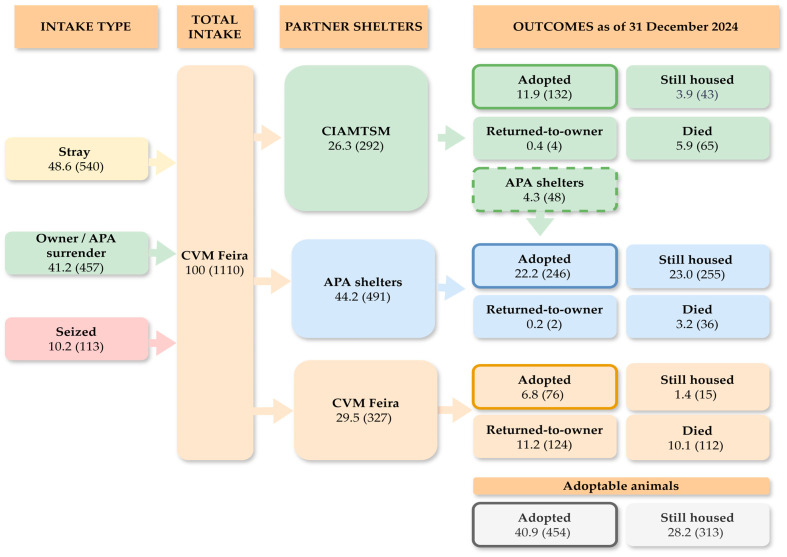
Overall flow of intake and outcomes of dogs (*n* = 1110) in CVM Feira (2018–2024). Values are presented as % (*n*).

**Figure 3 animals-16-00141-f003:**
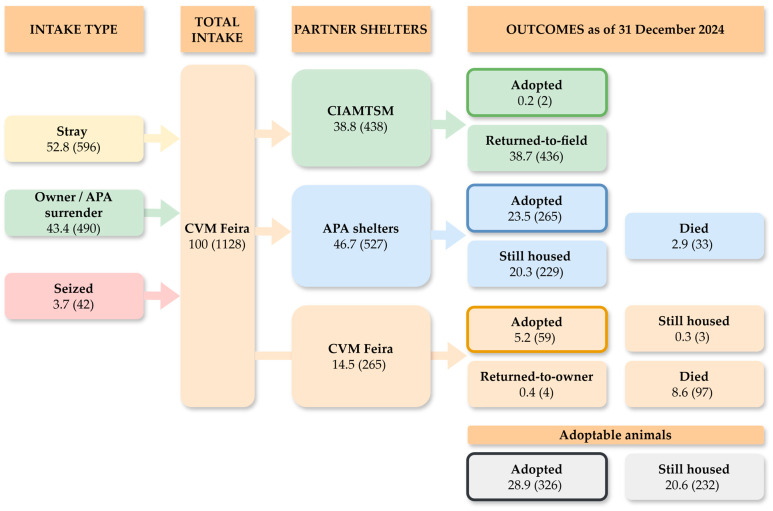
Overall flow of intake and outcomes of cats (*n* = 1128) in CVM Feira (2018–2024). Values are presented as % (*n*).

**Figure 4 animals-16-00141-f004:**
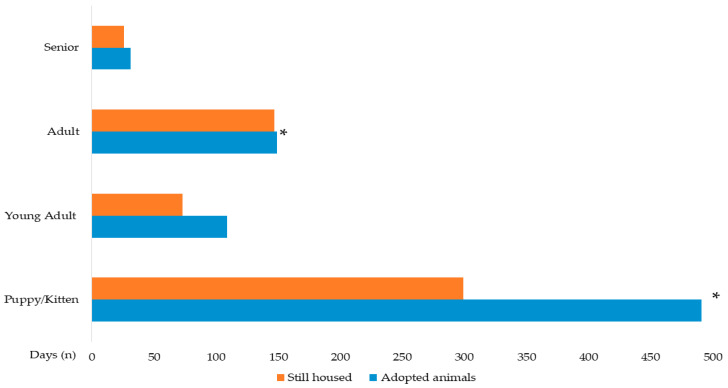
Age at intake of adopted animals (*n* = 780) and still housed animals (*n* = 545). * Denotes statistical significance (*p* < 0.05) in the Pearson’s Chi-Square test and Post hoc analyses.

**Figure 5 animals-16-00141-f005:**
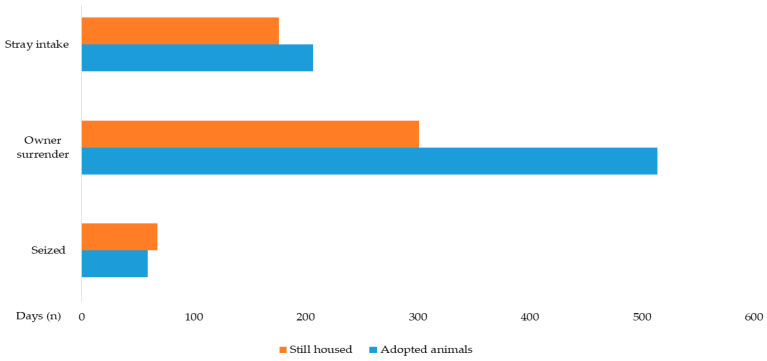
Intake type of adoptable animals (*n* = 1325), including adopted animals (*n* = 780) and still housed animals (*n* = 545). Statistical significance (*p* < 0.05) in the Pearson’s Chi-Square test and Post hoc analyses.

**Figure 6 animals-16-00141-f006:**
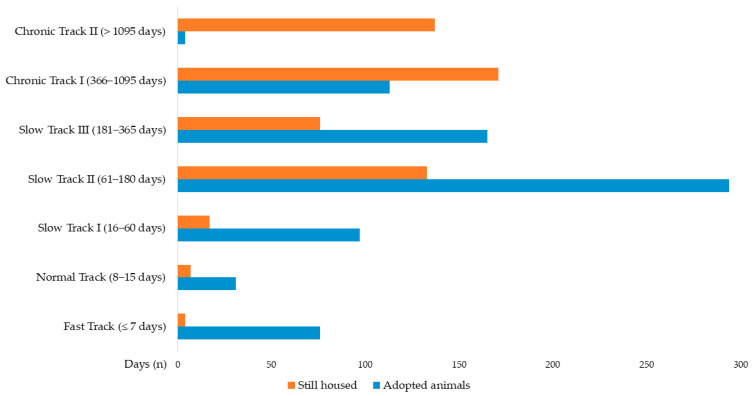
Length of Stay (LOS) of adoptable animals (*n* = 1325), including adopted animals (*n* = 780) and still housed animals (*n* = 545). Statistical significance (*p* < 0.05) in the Pearson’s Chi-Square test and Post hoc analyses.

**Figure 7 animals-16-00141-f007:**
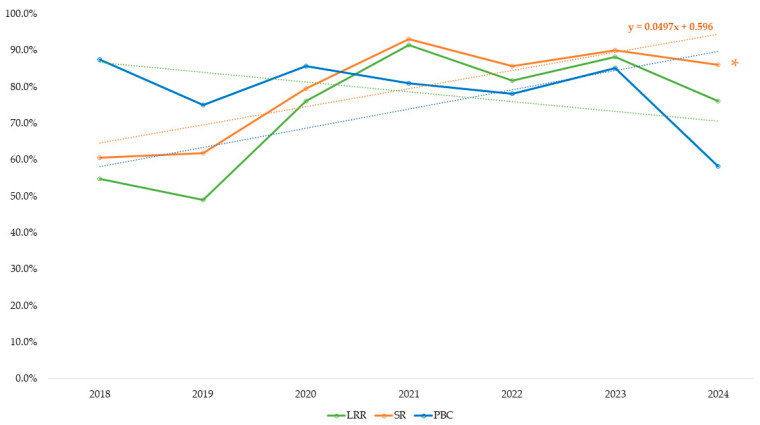
Trends in Live Release Rate (LRR), Save Rate (SR) and Population Balance Calculation (PBC) for dogs and cats (*n* = 2238) admitted to CVM Feira (2018–2024) (Shelter Animals Count, 2025). * Denotes statistical significance (*p* < 0.05) in the regression analysis.

**Table 1 animals-16-00141-t001:** Descriptive characteristics of the CVM Feira shelter intake population between 2018 and 2024 (*n* = 2238). Values are presented as % (*n*).

Characteristics	Dogs	Cats	Total
Demographic characteristics			
Species	49.6 (1110)	50.4 (1128)	100 (2238)
Sex			
Female	43.8 (486)	59.7 (673)	51.8 (1159)
Male	56.2 (624)	40.3 (455)	48.2 (1079)
Age at intake			
Puppy/Kitten (≤6 months)	41.4 (460)	42.6 (480)	42.0 (940)
Young adult (>6 months to ≤2 years)	11.9 (132)	27.3 (308)	19.7 (440)
Adult (>2 to ≤8 years)	36.4 (404)	28.0 (316)	32.3 (720)
Senior (>8 years)	10.3 (114)	2.1 (24)	6.2 (138)
Breed group			
Mixed breed	87.3 (969)	98.3 (1109)	92.9 (2078)
Recognized breed	11.3 (125)	1.7 (19)	6.4 (144)
Potentially dangerous breed	1.4 (16)	-	0.7 (16)
Body size (*dogs only*)			
Small	56.7 (629)	-	-
Medium	27.8 (309)	-	-
Large	15.5 (172)	-	-
Administrative characteristics			
Microchipped at intake			
No	81.4 (903)	99.6 (1123)	90.5 (2026)
Yes	18.6 (207)	0.4 (5)	9.5 (212)
Intake Type			
Stray	48.6 (540)	52.8 (596)	50.8 (1136)
Owner/APA Surrender	41.2 (457)	43.4 (490)	42.3 (947)
Seized	10.2 (113)	3.7 (42)	6.9 (155)

**Table 2 animals-16-00141-t002:** Logistic regression models describing the probability of being adopted, based on animal characteristics for adoptable dogs and cats (*n* = 1325), dogs (*n* = 767), and cats (*n* = 558) at CVM Feira.

Characteristics	Odds Ratio (95% CI)	*p*-Value
**Dogs**		
**Sex ^a^** **Age at intake**	0.664 (0.497–0.887)	0.006 *
Puppy (≤6 months)	Reference	-
Young adult (>6 months–2 years)	0.811 (0.525–1.253)	0.346
Adult (>2–8 years)	0.650 (0.466–0.907)	0.011 *
Senior (8+ years)	0.793 (0.424–1.484)	0.468
**Intake type**		
Seized	Reference	-
Owner/APA Surrender	2.233 (1.396–3.571)	<0.001 *
Stray	1.268 (0.794–2.024)	0.32
**Cats**		
**Age at intake**		
Kitten (≤6 months)	Reference	-
Young adult (>6 months–2 years)	1.048 (0.629–1.747)	0.856
Adult (>2–8 years)	0.463 (0.276–0.777)	0.004 *
Senior (8+ years)	0.456 (0.142–1.461)	0.186
**Intake type**		
Seized	Reference	-
Owner APA Surrender	1.965 (0.999–3.866)	0.050
Stray	1.666 (0.727–3.816)	0.228
**Dogs and Cats**		
**LOS** Fast Track (≤7 days)	Reference	-
Normal Track (8–15 days)	0.233 (0.064–0.853)	0.028 *
Slow Track I (16–60 days)	0.300 (0.097–0.929)	0.037 *
Slow Track II (61–180 days)	0.116 (0.042–0.325)	<0.001 *
Slow Track III (181–365 days)	0.114 (0.040–0.324)	<0.001 *
Chronic Track I (366–1095 days)	0.035 (0.012–0.098)	<0.001 *
Chronic Track II (>1095 days)	0.002 (0.000–0.006)	<0.001 *

* Denotes statistical significance (*p* < 0.05). ^a^ Females were coded as the reference category.

## Data Availability

The data that support the findings of this study are available from the corresponding author upon reasonable request, respecting confidentiality agreements with the municipal shelter and partner institutions.

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
