# Peer review of "Trends in Intake, Length of Stay and Outcome Data in a Portuguese Animal Shelter Between 2018 and 2024"

_animals, 2026, doi:10.3390/ani16010141_

Round 1
Reviewer 1 Report
Comments and Suggestions for Authors
I appreciate the authors’ efforts in producing a comprehensive and carefully executed study on this topic.
The manuscript is understandable, well-organised, and methodologically coherent, with the results section presented in an appropriate manner. I particularly appreciate the clarity of Figures 2 and 3, which effectively illustrate the overall intake and outcome of dogs and cats. I do not have any comments that would fundamentally alter the manuscript; below, I list only a few minor points.
Introduction: Given that the authors did not focus on a broader range of shelters across Portugal but instead relied on a relatively limited dataset derived from a single system, I recommend moderating the wording in line 113, where the text suggests placing the study’s findings in a global context — a claim that may be too strong. I therefore suggest rephrasing the end of the Introduction accordingly.
Materials and Methods: Line 171 – The categorisation of dogs into small, medium, and large should be further specified, for example by providing the height at the withers used for each category, so that readers can clearly understand the criteria underlying these classifications.
Regarding the inclusion of animals in the shelter system, I would also be interested to know how data of animals entering the shelter more than once were handled, and whether such cases were identifiable in the dataset. Based on my experience from shelters in the Czech Republic, repeated admissions following unsuccessful adoptions are relatively common, with some animals returning multiple times before being permanently rehomed. It would be important to clarify how such cases were incorporated into the dataset.
From the reader’s perspective, the methodological section would benefit from additional information about the individual facilities. At present, it is unclear what the shelter capacities are, how animals are housed, the duration of the quarantine period, and what veterinary or procedural practices are followed (e.g., behavioural assessment protocols). These operational aspects, along with adoption management strategies (such as activities aimed at increasing adoption likelihood and the adoption process itself), can substantially influence overall outcomes. Could the authors provide these details to give readers a more comprehensive understanding of how the shelters are operated? Even seemingly minor internal practices may significantly affect the results.
Discussion: In the Results section (line 294), the authors state that in 2018–2019 most admitted animals were dogs, whereas in 2020 the intake of cats increased. Could the authors elaborate in the Discussion on the factors that may explain this shift? Additionally, in lines 400–401, the authors mention that large dogs and dogs of “dangerous breeds” had a short length of stay (LOS). In practice, data from other countries often show the opposite trend, with such dogs being more difficult to rehome, as supported by various studies. Could the authors provide further explanation for this finding?
Line 462 – The authors state that the intake of animals in critical condition influenced the mortality rate. As the study did not investigate causes of mortality nor formally analyse them, this remains speculative. I would recommend either moderating this claim or removing it, as it is not directly supported by the data presented.
L490 - Line 490 – The authors report that juveniles had the longest median LOS and suggest that this may reflect the time required for growth, medical care, etc. From this perspective, it would have been more appropriate to subdivide the juvenile group into two categories (e.g., 0–3 months and 3–6 months), as the need for growth-related care would apply only to the youngest animals that still required time to remain with their mother until weaning. I therefore recommend rephrasing this statement and avoiding references to growth. Vaccination and neutering may also be relevant considerations; if most juveniles were close to 6 months of age, they would typically already be neutered and vaccinated, and thus the explanation provided may not apply to the majority of the group.Author Response
|
1. Summary |
|
|
|
Thank you very much for taking the time to review our manuscript and for your positive assessment of the study’s quality, clarity, and methodological coherence. We are grateful for your constructive minor comments, which have helped us improve the manuscript. All changes are implemented in the revised version and highlighted with track changes.
|
||
|
2. Point-by-point response to Comments and Suggestions for Authors |
||
|
Comments 1: “Introduction: Given that the authors did not focus on a broader range of shelters across Portugal but instead relied on a relatively limited dataset derived from a single system, I recommend moderating the wording in line 113, where the text suggests placing the study’s findings in a global context — a claim that may be too strong. I therefore suggest rephrasing the end of the Introduction accordingly.” |
||
|
Response 1: Thank you for this helpful suggestion. We agree that the original wording could overstate the global generalizability of our findings given that the data derive from a single municipal shelter system. We have therefore moderated the final sentence of the Introduction to better reflect the scope of the study. Change implemented: · Introduction (page 3, lines 113–114): o Previous text (end of paragraph): “while offering comparative insights relevant to broader European and global contexts.” o Revised text: “while offering insights relevant to comparable European shelter systems.” |
||
|
Comments 2: “Materials and Methods: Line 171 – The categorisation of dogs into small, medium, and large should be further specified, for example by providing the height at the withers used for each category, so that readers can clearly understand the criteria underlying these classifications.” |
||
|
Response 2: Thank you for pointing out the need to clarify the criteria used for body size categorization. In the original submission, the manuscript only mentioned “small, medium, large” with a generic reference to national legislation, which did not provide sufficient detail for international readers. In the revised version, we explicitly state that body size classification was based on approximate height at the withers and clarify that these categories are aligned with Fédération Cynologique Internationale (FCI) agility regulations. Dogs were assigned to three size groups: small (<35 cm), medium (35–<48 cm), and large (≥48 cm). The “medium” category in this study pragmatically aggregates the FCI medium and intermediate classes into a single group for analytical purposes. Using internationally established height thresholds facilitates comparison with other studies and kennel-based classifications. These changes are now described in the Materials and Methods.
Changes implemented:
o Previous text: o Revised text (page 5, lines 199-200): “[30]; and Body size (dogs only: small, medium, large) [31]. o Revised text (pages 5-6, lines 224-230): “Operational definition of body size: Body size classification was applied to dogs only, which were assigned to three size groups, following the height-at-the-withers categories adapted from FCI agility regulations: small (<35 cm), medium (35–<48 cm), and large (≥48 cm). The “medium” category in this study therefore aggregates the FCI medium (35–<43 cm) and intermediate (43–<48 cm) classes into a single group for analytical purposes [32].”
Comments 3: “Regarding the inclusion of animals in the shelter system, I would also be interested to know how data of animals entering the shelter more than once were handled, and whether such cases were identifiable in the dataset. Based on my experience from shelters in the Czech Republic, repeated admissions following unsuccessful adoptions are relatively common, with some animals returning multiple times before being permanently rehomed. It would be important to clarify how such cases were incorporated into the dataset.” Response 3: Thank you for raising this important point. In this retrospective study, each intake event was recorded and analyzed as an independent record, and the internal database was not designed to systematically flag repeated admissions of the same individual (for example, following unsuccessful adoptions or returns). As a result, potential recidivism could not be reliably identified or quantified, and cases of multiple admissions of the same animal may be present but are not distinguishable in the dataset. Given the municipal–intermunicipal structure of the system, in which most animals are transferred from CVM Feira to CIAMTSM and APA partners for long-term housing and subsequent adoption, any post-adoption returns are more likely to occur and be recorded at those downstream facilities rather than as repeat intakes at CVM Feira. This limitation is now explicitly acknowledged in the Methods, and recidivism was not analyzed as a separate outcome in the present study. Changes implemented: · Materials and Methods – Section 2.1. Study Design and Population (page 4, lines 148-151): o New paragraph added: “Each intake event was recorded independently under a unique identifier. The internal database was not designed to systematically flag repeated admissions of the same animal (e.g., following unsuccessful adoptions or returns), and therefore recidivism could not be reliably identified or quantified in this analysis.” · Discussion – Section 4.8. Strengths and Limitations (page 21, lines 764-770): o New sentence added: “Data on repeated admissions (recidivism) were not systematically captured in the database, preventing formal assessment of readmission rates. Although the structure of the municipal–intermunicipal system suggests that most relinquishments, if they occur, would be registered at downstream CIAMTSM and APA partner shelters rather than at CVM Feira, the absence of structured recidivism data remains a methodological limitation and an area for future investigation.”
Comments 4: “From the reader’s perspective, the methodological section would benefit from additional information about the individual facilities. At present, it is unclear what the shelter capacities are, how animals are housed, the duration of the quarantine period, and what veterinary or procedural practices are followed (e.g., behavioural assessment protocols). These operational aspects, along with adoption management strategies (such as activities aimed at increasing adoption likelihood and the adoption process itself), can substantially influence overall outcomes. Could the authors provide these details to give readers a more comprehensive understanding of how the shelters are operated? Even seemingly minor internal practices may significantly affect the results.” Response 4: Thank you for this suggestion. In the revised manuscript, we added a dedicated subsection describing the main facilities involved (CVM Feira, CIAMTSM, and APA partners), their housing capacities, and how animals are managed operationally, including basic housing routines, behavioral observations, and core veterinary and pre-adoption procedures. These additions provide readers with a clearer picture of the shelter context in which intakes, LOS, and outcomes occur, and help to situate the findings within established shelter care guidelines. Change implemented: · Materials and Methods – New subsection 2.1.1. Shelter Facilities and Operational Practices (page 4, lines 153-172): o The following text was added: “The CVM Feira functions as an Official Municipal Animal Shelter (CRO) and operates in close collaboration with the Official Intermunicipal Animal Shelter of the Association of Municipalities of Terras de Santa Maria (CIAMTSM) and several non-profit animal protection associations (APA). CVM Feira has 12 dog kennels and a small core team (one veterinarian, one caretaker and one administrative staff member), which limits on-site housing capacity. Animals requiring longer stays or additional space are transferred to CIAMTSM, whose main facility comprises 51 dog kennels distributed across four wings, three semi-circular quarantine kennels, an additional 10-kennel wing, and 2 dog pack parks for group housing of dogs, which became operational in 2021.​ Newly admitted dogs are placed in clean, disinfected kennels, and animals are periodically relocated between kennels to allow thorough cleaning and disinfection before receiving new intakes. Behavioral assessment is based on veterinarian and caretaker observations during routine handling and is used to inform housing and adoption counselling. According to local protocols, animals adopted from the municipal system are routinely dewormed internally and externally, vaccinated according to age, identified with a microchip (if not already identified), and sterilized (spayed or castrated) when clinically appropriate, either before adoption or shortly thereafter.”
Comments 5: “Discussion: In the Results section (line 294), the authors state that in 2018–2019 most admitted animals were dogs, whereas in 2020 the intake of cats increased. Could the authors elaborate in the Discussion on the factors that may explain this shift?” Response 5: Thank you for this observation. We agree that the change in species composition over time warrants further discussion. In the revised Discussion, we now note that the increase in cat intake from 2020 onwards likely reflects the national implementation and expansion of trap–neuter–return (TNR) programmes for free-roaming cats and the formal registration of community colonies, which have increased the visibility of cats in official records. At the local level, strengthened partnerships with CIAMTSM and animal protection associations (e.g., AANIFEIRA) have facilitated the intake and placement of cats and senior animals in the absence of a dedicated municipal cattery. Together, these factors likely contributed to the higher proportion of cats entering and being recorded within the municipal system during the study period. Change implemented: · Discussion - 4.3. Species-Specific Intake Trends and Management Implications (page 19, lines 642-650): o Added new section: “The transition from predominantly dog intakes in 2018–2019 to an increasing proportion of cats from 2020 onwards likely reflects both national and local developments. National implementation and expansion of trap–neuter–return (TNR) programmes and the formal registration of community cat colonies increased the visibility of free-roaming cats within official records. Locally, partnerships with the CIAMTSM and APA shelters have facilitated the intake and placement of cats and senior animals, partially compensating for the absence of a dedicated municipal cattery. These factors together help explain the growing representation of cats in the municipal shelter system over time.”
Comments 6: “Additionally, in lines 400–401, the authors mention that large dogs and dogs of “dangerous breeds” had a short length of stay (LOS). In practice, data from other countries often show the opposite trend, with such dogs being more difficult to rehome, as supported by various studies. Could the authors provide further explanation for this finding?” Response 6: Thank you for drawing attention to this apparent discrepancy with findings from other countries, where large dogs and dogs labeled as restricted or “dangerous” breeds are often more difficult to rehome and tend to have longer LOS. In our study, the analysis of predictors of adoption was restricted to a subsample of 1,325 “adoption-track” animals with only two outcomes (adopted vs still housed), so LOS in this context reflects time to adoption among animals considered eligible for placement and does not include non-live outcomes or RTO/RTF cases. In the revised Discussion, under the “Breed group” and “Body size” subsections, we now clarify that: (i) the number of dogs classified as “potentially dangerous breeds” under Portuguese legislation was small; (ii) shorter LOS for these dogs and for large dogs likely reflects a combination of adopter interest in certain recognized or restricted breeds and targeted promotion, transfer, and adoption counselling efforts by the municipal shelter network and APA partners; and (iii) local capacity constraints and regional adopter preferences may contribute to faster placement of some large or legally restricted dogs, so direct extrapolation to other national contexts should be made with caution. Changes implemented: · Discussion – Section 4.2. Predictors of Adoption, Breed group subsection (pages 18-19, lines 616-634): o Revised text: “Breed group: Animals of recognized breeds had a shorter median LOS (19 days; IQR: 195 days) compared to mixed-breed animals (91 days; IQR: 242 days), which is consistent with studies suggesting adopter preferences for specific, recognizable breeds and for animals perceived as having more predictable adult characteristics, such as size, temperament, and care requirements. In this shelter system, breed information was communicated to adopters only at an aggregated level (mixed-breed vs recognized breed vs potentially dangerous breed), but even this level of classification may shape adopter expectations. Dogs classified as potentially dangerous breeds under Portuguese legislation also showed relatively short LOS among adoption-track animals (median 28 days). This finding diverges from reports in other countries, where restricted breeds are often more difficult to rehome and have longer LOS. In our context, it should be interpreted considering the small number of dogs in this category, which limits statistical robustness, and of targeted efforts by the municipal shelter network and APA partners to promote these dogs through focused communication, transfers, and adoption counselling. Local adopter preferences and housing conditions may further modulate demand for specific breed types, making direct comparison with other national settings difficult.” · Discussion – Section 4.8. Strengths and Limitations (page 21, lines 748-753): o Revised text: “Likewise, breed group classification followed a pragmatic, three-category approach based on phenotypic assessment rather than objective pedigree or genetic methods. These methodological choices, together with context-specific management strategies and adoption promotion efforts, may have influenced the observed associations between body size, breed group and LOS and should be considered when interpreting the study findings.”
Comments 7: “Line 462 – The authors state that the intake of animals in critical condition influenced the mortality rate. As the study did not investigate causes of mortality nor formally analyse them, this remains speculative. I would recommend either moderating this claim or removing it, as it is not directly supported by the data presented.” Response 7: We agree that the original statement was too strong given that we did not conduct a formal analysis of mortality causes. We have therefore moderated the wording to explicitly acknowledge its speculative nature. Changes implemented: · Discussion, Section 4.1. Ethical Success and Population Management Challenges in a « No-Kill » Framework (page 17, lines 547-549): o Previous text: o Revised text:
Comments 8: “L490 - Line 490 – The authors report that juveniles had the longest median LOS and suggest that this may reflect the time required for growth, medical care, etc. From this perspective, it would have been more appropriate to subdivide the juvenile group into two categories (e.g., 0–3 months and 3–6 months), as the need for growth-related care would apply only to the youngest animals that still required time to remain with their mother until weaning. I therefore recommend rephrasing this statement and avoiding references to growth. Vaccination and neutering may also be relevant considerations; if most juveniles were close to 6 months of age, they would typically already be neutered and vaccinated, and thus the explanation provided may not apply to the majority of the group.” Response 8: Thank you very much for these insightful comments. The juvenile age band (0–6 months) used in this study was deliberately aligned with age categories widely applied in shelter research, including Powell et al. (2021), to facilitate comparability across studies. We fully agree that, conceptually, subdividing this juvenile group into narrower bands (e.g., 0–3 and 3–6 months) would provide a more nuanced understanding of LOS, given the different biological and management needs of neonates versus older juveniles. In the present analysis, however, all models were specified and run using the 0–6 month juvenile band, and we therefore retained this categorization to preserve internal consistency, while refining the interpretation of juvenile LOS as you suggested. In line with your recommendation, we have rephrased the relevant Discussion section to remove references to “growth” as the main explanation and now focus on operational factors, including age‑ and weight‑dependent pre‑adoption procedures (vaccination, deworming, microchipping, and sterilization), foster availability for very young animals, and the broad 0–6 month intake range when interpreting the longer LOS observed for juveniles. We also note that adopter preferences within the juvenile band may play a role, as very young puppies and kittens are often favored over slightly older juveniles, which is consistent with previous studies showing a general preference for younger animals in shelter adoptions. Finally, we explicitly acknowledge in the Strengths and Limitations section that the use of a single 0–6 month juvenile category, following Powell et al., is a methodological limitation and that future work in this context should consider subdividing this group into narrower age ranges to better capture differences in care needs and adoption readiness. Changes implemented:
o Previous text: “…juveniles had the longest median LOS… reflecting the time required for growth, medical care, etc.” o Revised text: “Age at intake: The bias against adult animals (dogs OR = 0.650, cats OR = 0.463) is a key finding, consistent with existing literature and with age categories commonly used in shelter studies. Despite their higher odds of adoption, puppies and kittens had the longest median LOS (149 days; IQR 209 days), which likely reflects operational factors rather than lack of adopter interest. The need to complete age‑ and weight‑dependent pre‑adoption procedures (vaccination, deworming, microchipping, and sterilization), together with age‑appropriate vaccination schedules and postponed sterilization until animals reach a suitable weight or age, can delay adoption readiness and prolong shelter stays for juveniles. This juvenile category also covers a broad intake age range (0–6 months), so some animals may enter as neonates requiring intensive care or foster placement, while others arrive closer to typical adoption age, which should be kept in mind when interpreting the mean LOS of 149 days for this group. Adopter preferences within this band may also contribute, as very young puppies and kittens are often favored over slightly older juveniles, in line with previous work showing a general preference for younger shelter animals. In contrast, many young adults typically enter already meeting core adoption requirements (or can be prepared more rapidly), helping to explain their shorter LOS, whereas older animals, particularly senior cats, continue to face adoption barriers linked to health and longevity concerns.” · Discussion – Section 4.8. Strengths and Limitations (page 21, lines 744-748): o Added sentence: “Age at intake was categorized using a 0–6 month juvenile band [28], to facilitate comparability with previous shelter research; however, not subdividing this group into narrower age ranges (e.g., 0–3 vs 3–6 months) may mask important differences in care needs, adopter preferences and adoption readiness within the juvenile category and should be considered a methodological limitation.” |
||
We thank you again for the thoughtful and constructive comments, which have helped to clarify the methods, contextualize the results, and improve the overall quality of the manuscript.
December 17, 2025

Reviewer 2 Report
Comments and Suggestions for Authors
Thank you for the opportunity to review this very well-done study. Below, I provide a few suggestions to further improve its quality.
- 136-140: perhaps note that intaking RTF animals does affect shelter stats
- 140-144: consider discussing the implications of classifying found strays brought to the shelter as OS instead of stray
- 195-244: are transfers out, as described in l.130, included in live outcomes? If so, I don’t see them described here.
- 230-233: consider describing the pros and cons of the different approaches to calculating LRR, especially as some readers may not be familiar with these nuances. While true that using intakes as the LRR denominator may “underestimate the shelter’s live release performance by including animals still housed at the end of the reporting period” and therefore arriving at a lower percentage, I believe we should be less concerned about shelter “performance” and more concerned about shelter warehousing of animals to achieve better-looking statistics. Reporting outcomes only as a % of all outcomes can mask detriment to animal welfare that occurs by keeping long-stay animals in care even when they are likely unadoptable. While I am by no means suggesting that the shelter involved in this study warehouses animals, and I recognize that various methods of calculating LRR are in accepted use in the sheltering community, I do think that a few lines about the implications of different approaches to the calculation would be useful either here or in the Discussion.
- Related: consider whether, then, reporting save rate—which does use intakes as the denominator—skews toward overestimating shelter performance in some regards, and whether it makes more sense either to use or not use the intake data wholesale rather than including it in one calculation and excluding it from the other.
- 237-239: similarly, I feel it is important to note that while save rate does indeed tell us the proportion of animals entering the shelter that did not experience a non-live outcome, it does not tell us what proportion of those animals had an outcome of any kind—thereby not accounting for animals who remain in the shelter. Since LOS is an important variable in this study, a bit more discussion here seems of value.
- 284 is sterilization status at intake available to add to this table, especially given the paper’s themes around TNR?
- Fig. 1: consider post hoc analysis to identify between what years the significant omnibus differences occurred
- In Table 2, the OR for Senior is given with a comma instead of a decimal point.
- 370: were animals still in shelter at the conclusion of the study period included in LOS analysis, or only those animals with outcomes (and thus final LOS)? L. 411 seems to indicate that the still-in-shelter animals were included; I am wondering what the implications for the findings might have been if using only completed LOS.
- 400-402: might it be worth separately calculating LOS to live outcome and LOS to non-live outcome? Potentially dangerous breeds and large dogs, I suspect, had shorter LOS due to euthanasia decisions being made sooner—rather than because they got snapped up for adoption.
- 437 “authors’”
- 454 “ethically”
- 482-483: is it not also possible that the causality is reversed, either instead or in addition: lower adoptability from the outset increases LOS?
- 489: mean LOS of 149 days for puppies and kittens represents 5 months if they enter as absolute newborns, but most puppies and kittens would be considered “ready to go” (vaccinated, sterilized, and socialized) well before this age. Does the shelter delay sterilization? Are very young kittens kept in the shelter or diverted to alternate (e.g., foster) placement?
- 502-505: what happens if you more clearly compare apples to apples by removing the community cats from this dog-v.-cat analysis?
- 508-510: how was breed determined, and then conveyed to potential adopters? These factors potentially influence how breed was associated with LOS.
- 552 should be “restraints” or more accurately “constraints”
- 564-568 duplicate sentence
- 573-575 yes, though consider also (and consider noting) that owners who microchip their animals may also be more likely to be proactive in locating them if they go missing, thereby increasing RTO among this subset of animals
Author Response
|
1. Summary |
|
|
|
Thank you very much for taking the time to review this manuscript and for the detailed, thoughtful suggestions to further improve the manuscript. Please find the detailed responses below and the corresponding revisions/corrections highlighted/in track changes in the re-submitted files.
|
||
|
2. Point-by-point response to Comments and Suggestions for Authors |
||
|
Comments 1: “136–140: perhaps note that intaking RTF animals does affect shelter stats”. |
||
|
Response 1: Thank you for this suggestion. We agree that explicitly noting the impact of including RTF animals on shelter statistics is important. We have added a clarifying sentence to the Materials and Methods. Change implemented: · Materials and Methods - Section 2.1. Study Design and Population (pages 3-4, lines 140-143): o New sentence added after describing the inclusion of RTF animals as both intake and outcome events: “It is acknowledged that including RTF animals in intake statistics may influence shelter performance metrics, as these animals are not available for adoption and have predetermined outcomes.”
|
||
|
Comments 2: “140–144: consider discussing the implications of classifying found strays brought to the shelter as OS instead of stray.” |
||
|
Response 2: We appreciate this nuanced point. we have kept the detailed operational definition of Owner/APA surrenders in the Materials and Methods, clarifying that animals initially found as strays by citizens, municipal services or APA and later formally transferred to the shelter are coded in this category. We have additionally added a short paragraph in the Discussion explicitly noting that this choice may underestimate the proportion of true stray intakes and overestimate owner‑initiated surrenders when compared with shelters that use stricter SAC definitions, while still reflecting the operational reality of animals formally transferred through community partnerships. Change implemented: · Discussion - 4.3. Species-Specific Intake Trends and Management Implications, (page 19, lines 661-670) o Added paragraph: “It is also important to consider how local intake classification practices influence the interpretation of these findings. In this municipal–intermunicipal system, animals initially found as strays by citizens or APA and temporarily kept under their care, or those registered for external adoption, were formally surrendered to CVM Feira for veterinary procedures and subsequently adopted, and were therefore coded as Owner or APA surrenders at intake. This approach may affect comparability with shelters using stricter SAC definitions, as it can underestimate the proportion of true stray intakes and overestimate owner‑initiated surrenders, even though it accurately reflects the operational reality of animals formally transferred through community partnerships [20].”
Comments 3: “195–244: are transfers out, as described in l.130, included in live outcomes? If so, I don’t see them described here”. Response 3: Thank you for highlighting this omission. We have clarified how transfers to partner facilities were treated in outcome classification to avoid double counting. Change implemented: · Materials and Methods - Section 2.3.2. Outcome Variables and Performance Metrics (pages 6-7, lines 265-269): o New text integrated: “Transfers out to partner shelters (CIAMTSM and APA) were tracked throughout the animal’s trajectory. Only final outcomes (adoption, RTO, RTF, died, or still housed) at the last shelter where the animal resided were included in outcome calculations to avoid double counting.”
Comments 4: “230-233: consider describing the pros and cons of the different approaches to calculating LRR, especially as some readers may not be familiar with these nuances. While true that using intakes as the LRR denominator may “underestimate the shelter’s live release performance by including animals still housed at the end of the reporting period” and therefore arriving at a lower percentage, I believe we should be less concerned about shelter “performance” and more concerned about shelter warehousing of animals to achieve better-looking statistics. Reporting outcomes only as a % of all outcomes can mask detriment to animal welfare that occurs by keeping long-stay animals in care even when they are likely unadoptable. While I am by no means suggesting that the shelter involved in this study warehouses animals, and I recognize that various methods of calculating LRR are in accepted use in the sheltering community, I do think that a few lines about the implications of different approaches to the calculation would be useful either here or in the Discussion.” Response 4: We are very grateful for this thoughtful and critical reflection on LRR calculation and its ethical implications. In the revised manuscript, we have expanded the Materials and Methods section to explicitly present both the outcome‑based definition of LRR used by Shelter Animals Count (live outcomes ÷ total outcomes, excluding animals still housed) and the alternative intake‑based formulation (live outcomes ÷ total intake), and to clarify why the SAC outcome‑based definition was adopted in this study for purposes of international comparability. We also added a short paragraph in the Discussion explaining that outcome‑based LRR and intake‑based metrics (SR and PBC) capture different but complementary aspects of shelter functioning: outcome‑based LRR focuses on the fate of animals with completed outcomes, whereas intake‑based measures emphasize population flow and can reveal the accumulation of long‑stay animals that may be associated with welfare concerns. We now explicitly state that these indicators should be interpreted together when assessing both life‑saving performance and the risk of “warehousing” animals in long‑term care. Changes implemented: · Materials and Methods - Section 2.3.2. Outcome Variables and Performance Metrics (Live Release Rate definition) (page 7, lines 289-299): o Previous text: “Live Release Rate (LRR): A key performance indicator calculated as the proportion of Live Outcomes (adoptions, RTO and RTF) divided by total outcomes (live and non-live outcomes), excluding animals stillhoused, following SAC methodology [12]. This approach was chosen instead of calculating LRR based on intakes, as the latter may underestimate the shelter’s live release performance by including animals still housed at the end of the reporting period. A high LRR is widely recognized as a primary indicator of a shelter's commitment to saving lives and is a common benchmark in the animal welfare community [29].” o Revised text: “Live Release Rate (LRR): LRR was calculated following the outcome‑based definition used by Shelter Animals Count (SAC), as the proportion of live outcomes (adoptions, RTO and RTF) divided by total outcomes (live and non‑live outcomes), excluding animals still housed at the end of the reporting period. This outcome‑based approach is widely adopted in the sheltering community and facilitates comparison with existing benchmarks. An alternative, intake‑based formulation defines LRR as the proportion of animals leaving alive divided by total intake (as used, for example, in the ASPCA LRR), which places greater emphasis on population flow and the contribution of new admissions to live outcomes.” · Discussion – Section 4.1. Ethical Success and Population Management Challenges in a “No-Kill” Framework (page 17, lines 560-570): o New paragraph added : “In this study, we adopted the SAC outcome‑based LRR to ensure consistency with international reporting standards, but we interpret it together with intake‑based indicators such as SR and PBC. Different formulations of LRR capture distinct but complementary aspects of shelter functioning. Outcome‑based LRR, as used here, focuses on the fate of animals with completed outcomes and is useful for benchmarking life‑saving performance, whereas intake‑based metrics such as SR and PBC place greater emphasis on population flow and can highlight the accumulation of long‑stay animals (‘warehousing’) that may not be apparent when only outcome‑based percentages are considered. Interpreting these indicators together is therefore essential when evaluating both ethical commitments to saving lives and the welfare implications of prolonged confinement [11].”
Comments 5: “Related: consider whether, then, reporting save rate—which does use intakes as the denominator—skews toward overestimating shelter performance in some regards, and whether it makes more sense either to use or not use the intake data wholesale rather than including it in one calculation and excluding it from the other.” Responses 5: Thank you for emphasizing this conceptual point. We agree that SR, although intake-based, does not indicate what proportion of animals actually left the shelter and can therefore overestimate “performance” if interpreted in isolation, especially when many animals remain in long‑term care. In the revised Materials and Methods, we have expanded the SR definition to state explicitly that SR reflects the proportion of animals that avoided non‑live outcomes but does not account for animals who remain in the shelter without a final outcome, and we link this directly to the role of LOS in our analyses. In addition, the new paragraph added to the Discussion explains that outcome‑based LRR and intake‑based metrics such as SR and PBC capture different but complementary aspects of shelter functioning and should be interpreted together when assessing both life‑saving capacity and the risk of warehousing animals in long‑term care. Changes implemented: · Materials and Methods - Section 2.3.2. Outcome Variables and Performance Metrics (Save Rate definition) (page 7, lines 302-308): o Previous text: “Save Rate (SR): Calculated as the total animal number of Live Outcomes divided by the Total Intake number. This metric reflects the proportion of animals entering the shelter that did not experience a non-live outcome, providing an intake-based measure of life-saving capacity [12];” o Revised text: “Save Rate (SR): SR was calculated as the total number of live outcomes divided by total intake, providing an intake‑based measure of the proportion of animals entering the shelter that did not experience a non‑live outcome. It should be noted that while SR reflects the proportion of animals that avoided non‑live outcomes, it does not indicate what proportion of animals had an outcome of any kind, thereby not accounting for animals who remain in long‑term shelter care.”
Comments 6: “237-239: similarly, I feel it is important to note that while save rate does indeed tell us the proportion of animals entering the shelter that did not experience a non-live outcome, it does not tell us what proportion of those animals had an outcome of any kind—thereby not accounting for animals who remain in the shelter. Since LOS is an important variable in this study, a bit more discussion here seems of value.” Response 6: As detailed in Response 5, the SR definition in Section 2.3.2 has been expanded to clarify its limitations and relation to LOS, and a new paragraph has been added to the Discussion explaining how outcome‑based LRR and intake‑based metrics (SR, PBC) should be interpreted jointly.
Comments 7: “284 is sterilization status at intake available to add to this table, especially given the paper’s themes around TNR?” Response 7: We appreciate this suggestion. Unfortunately, sterilization status at intake was not consistently recorded for all animals in the database used for this analysis, particularly for animals transferred from partner facilities or with incomplete historical records. We therefore could not reliably include this variable in Table 1. We now acknowledge this explicitly as a limitation. Change implemented: · Discussion – Section 4.8. Strengths and Limitations (page 21, lines 753-756): o Added new sentence:
Comments 8: “Fig. 1: consider post hoc analysis to identify between what years the significant omnibus differences occurred”. Response 8: Thank you for this suggestion. Figure 1 was intended to illustrate the temporal trend over the study period. Temporal trends were estimated using simple linear regression, fitting a line that minimizes the sum of squared residuals between observed and predicted values, following the approach described by Rodríguez et al. (2022). This was done because our interest was in examining trends over the entire five-year study period rather than year-to-year changes. Since this analysis does not involve an analysis of variance, the use of multiple comparison tests was not appropriate.
Comments 9: “In Table 2, the OR for Senior is given with a comma instead of a decimal point.” Response 9: Thank you for catching this formatting error. We have corrected the odds ratio for senior dogs in Table 2. Change implemented: · Results – Section 3.3. Predictors of Adoption (Table 2) (page 12, lines 424-425): o The formatting error was corrected: “0,793” → “0.793”.
Comments 10: “370: were animals still in shelter at the conclusion of the study period included in LOS analysis, or only those animals with outcomes (and thus final LOS)? L. 411 seems to indicate that the still-in-shelter animals were included; I am wondering what the implications for the findings might have been if using only completed LOS.” Response 10: We appreciate the opportunity to clarify this. Animals still housed at the end of the study period were indeed included in the LOS analysis, with LOS calculated up to 31 December 2024. We now state this explicitly and discuss the implication that LOS for some animals may be slightly overestimated relative to their eventual outcome. Change implemented: · Results - Section 3.3.4. Length of Stay as Primary Predictor (page 14, lines 460-464): o New text added: “Animals still housed at the end of the study period (December 31, 2024) were included in the LOS analysis, with their LOS calculated as the number of days from intake to the study end date. This approach provides a realistic representation of current shelter population dynamics, though it may slightly overestimate LOS for animals ultimately adopted shortly after the study period.”
Comments 11: “400-402: might it be worth separately calculating LOS to live outcome and LOS to non-live outcome? Potentially dangerous breeds and large dogs, I suspect, had shorter LOS due to euthanasia decisions being made sooner—rather than because they got snapped up for adoption.” Response 11: Thank you for this very pertinent suggestion. Conceptually, separating LOS to live outcomes from LOS to non‑live outcomes would indeed provide a clearer understanding of whether short LOS in specific subgroups reflects rapid adoption or earlier non‑live decisions. In our dataset, however, the number of dogs classified as potentially dangerous breeds under Portuguese legislation was very small (n = 16 in the full intake population, and only a subset of these were included in the adoption‑track analyses), and large dogs also represented a minority within the stratified models. This limited sample size substantially reduces the statistical robustness of any further stratification by outcome type and would make subgroup‑specific LOS estimates highly unstable and difficult to interpret with confidence.​ Moreover, implementing a full separation of LOS to live vs non‑live outcomes for all relevant subgroups would require re‑structuring several sections of the document. In response to your comment, we have therefore taken a more cautious interpretative approach: in the revised Discussion – Section 4.2. Predictors of Adoption, under the subsections Breed group and Body size, we explicitly acknowledge that short LOS for large dogs and for potentially dangerous breeds cannot be assumed to reflect uniformly rapid adoption, highlight the small numbers involved, and note that future studies with larger samples should formally distinguish LOS to live outcomes from LOS to non‑live outcomes to clarify these patterns.
Comments 12: “437 “authors’””. Response 12: Thank you for pointing out this language issue. The phrasing has been corrected from “to the authors knowledge” to “to the authors’ knowledge” in the revised manuscript (Discussion, page 16, line 523).
Comments 13: “454 “ethically””. Response 13: Thank you for drawing attention to this wording. The phrase has been corrected from “is ethical commendable” to “is ethically commendable” in the revised manuscript (Discussion – Section 4.1. Ethical Success and Population Management Challenges in a “No-Kill” Framework, page 16, line 540).
Comments 14: “482-483: is it not also possible that the causality is reversed, either instead or in addition: lower adoptability from the outset increases LOS?” Response 14: We fully agree. The direction of causality between LOS and adoptability is likely complex and potentially bidirectional. We have now explicitly acknowledged this in the Discussion. Change implemented: · Discussion – Section 4.2. Predictors of Adoption - Length of Stay (page 17, lines 581-587): o New text added stating that animals with inherently low adoptability may have longer LOS from the outset and that longitudinal behavioral and health data would be needed to clarified these pathways: “It is also possible that the causality operates in reverse, or bidirectionally: animals with inherently lower adoptability (e.g., due to age, health, or behavioral issues) may experience longer LOS from the outset, rather than LOS itself reducing adoptability. Clarifying these pathways would require longitudinal behavioral and health assessments, which were beyond the scope of this study [51].”
Comments 15: “489: mean LOS of 149 days for puppies and kittens represents 5 months if they enter as absolute newborns, but most puppies and kittens would be considered “ready to go” (vaccinated, sterilized, and socialized) well before this age. Does the shelter delay sterilization? Are very young kittens kept in the shelter or diverted to alternate (e.g., foster) placement?” Response 15: Thank you for these insightful comments regarding the interpretation of juvenile length of stay. In the revised manuscript, the previous explanation focused on “growth” has been reformulated with a more operationally oriented interpretation, emphasizing age- and weight-dependent pre-adoption procedures, foster availability, and the broad 0–6 month intake range. Change implemented:
“Age at intake: The bias against adult animals (dogs OR = 0.650, cats OR = 0.463) is a key finding, consistent with existing literature. Despite their higher odds of adoption, puppies and kittens had the longest median LOS (149 days, IQR 209 days), which likely reflects operational factors rather than lack of adopter interest. The need to complete age- and weight-dependent pre-adoption procedures (vaccination, deworming, microchipping, and sterilization), coupled with limited foster availability for very young animals, can delay adoption readiness and prolong shelter stays for juveniles. This juvenile category also covers a broad intake age range (0–6 months), so some animals may enter as neonates requiring intensive care or foster placement, while others arrive closer to typical adoption age; future studies should subdivide this group (e.g., 0–3 vs. 3–6 months) to capture these differences more precisely. The mean LOS of 149 days therefore warrants cautious interpretation: many puppies and kittens would, in principle, be ‘ready to go’ earlier under different sterilization policies, medical protocols, and foster capacity, and further research should examine how these operational factors influence LOS for neonatal and very young animals. In contrast, many young adults typically enter already meeting core adoption requirements (or can be prepared more rapidly), helping to explain their shorter LOS, whereas older animals, particularly senior cats, continue to face adoption barriers linked to health and longevity concerns [49,52].”
Comments 16: “502-505: what happens if you more clearly compare apples to apples by removing the community cats from this dog-v.-cat analysis? Response 16: Your question is pertinent. As species were not included as a predictor in the regression model of Predictors of Adoption, we remove the sentence, and had a sentence after in the discussion regarding LRR. Change implemented: · Previous text: “Species: Although species were not included as a predictor in the regression model, descriptive results showed that cats had a higher LRR (85.5%) than dogs (73.3%). This difference is largely explained by the Trap-Neuter-Return (RTF) program, which constitutes a live outcome independent of adoption [34,58]. In contrast, dogs depend almost entirely on adoption or return-to-owner (RTO) for their flow [49,53,54].“ · New subsection 4.3. Species-Specific Intake Trends and Management Implications (page 19, lines 651-660): “The higher LRR for cats (85.5%) compared to dogs (73.3%) is largely attributable to the Trap-Neuter-Return/Return-to-Field (TNR/RTF) program, which constituted a substantial proportion of cat live outcomes. When community cats returned to field are excluded and only adoption-track animals are considered, species-specific adoption dynamics still differ between dogs and cats, underscoring the need to interpret aggregate LRR with caution. This distinction is important when benchmarking shelter performance, as TNR/RTF programs represent a fundamentally different intervention pathway from traditional adoption, and outcome statistics for community cats and adoption-track populations should be reported separately whenever possible [34,62,63].”
Comments 17: “508-510: how was breed determined, and then conveyed to potential adopters? These factors potentially influence how breed was associated with LOS.” Response 17: Thank you for raising this important point. In our context, breed determination is primarily operational. Breed group classification was applied to dogs and cats using three categories: mixed-breed, recognized breed, and potentially dangerous breed (the latter defined according to Portuguese legislation). Under Portuguese law, animals without registration in the national studbook (Livro de Origens Português, LOP) are legally considered of undefined breed. As none of the animals in this dataset had studbook registration, all individuals without objective pedigree documentation or with uncertain breed identity were classified as mixed-breed, which represents the majority of the sample. Animals were classified as recognized breed when their phenotype and available documentation were consistent with a specific breed, based on the attending veterinarian’s visual assessment at intake and, when available, information provided by the owner or caretaker and breed data recorded in the national SIAC database for microchipped animals. International breed standards (FCI for dogs and FIFe for cats) were used as conceptual references for this visual assessment. Breed information was recorded and communicated to potential adopters only at this aggregated level (mixed-breed, recognized breed, potentially dangerous breed), and no breed‑by‑breed analyses were conducted. We have clarified these procedures in the Methods and now explicitly acknowledge in the Limitations that this pragmatic, visually based classification may affect the precision of breed-related analyses and should be considered when interpreting the association between breed group and LOS. Changes implemented: · Materials and Methods – Section 2.3.1 Intake Variables (page 5, lines 199-223):
“[30]; and Body size (dogs only: small, medium, large) [31]. Operational definition of breed group: Breed group classification was applied to dogs and cats using a three-category approach: mixed-breed, recognized breed, and potentially dangerous breed. Under Portuguese law (Decree‑Law No. 276/2001, consolidated version), “purebred” animals are those identified and registered with a genealogical record in the national studbook (Livro de Origens Português) [32]. In this dataset, none of the animals had studbook registration. For the purposes of this study, all animals without objective pedigree documentation or with uncertain breed identity were therefore grouped in the mixed-breed category, which corresponds to legally undefined-breed animals and represents the majority of the sample. Animals were classified as recognized breed when their phenotype and available documentation were consistent with a specific breed, based on the attending veterinarian’s visual assessment at intake and, when available, information provided by the owner or caretaker and breed data recorded in the national SIAC database for microchipped animals. International breed standards were used as conceptual references for this classification, in particular Fédération Cynologique Internationale (FCI) standards for dogs and Fédération Internationale Féline (FIFe) standards for cats, as applied in national kennel and feline registries. Breed information was recorded and communicated to potential adopters only at this aggregated level (mixed-breed, recognized breed, potentially dangerous breed), and no breed-by-breed analyses were conducted.”
“Likewise, breed group classification followed a pragmatic, three-category approach based on phenotypic assessment rather than objective pedigree or genetic methods. together with context-specific management strategies and adoption promotion efforts, may have influenced the observed associations between body size, breed group and LOS and should be considered when interpreting the study findings.”
o Added the following abbreviations: § “FCI - Fédération Cynologique Internationale” § “FIFe - Fédération Internationale Féline” § “LOP - Livro de Origens Português”
Comments 18: “552 should be “restraints” or more accurately “constraints” Response 18: Thank you for this careful observation. The wording has been updated. Change implemented: · Discussion, Section 4.6. COVID-19 pandemic and Socioeconomic Challenges (page 20, line 703): o The wording “socioeconomic restrains” was corrected to “socioeconomic constraints”.
Comments 19: “564-568 duplicate sentence” Response 19: We thank the reviewer for pointing this out. Change implemented: · Discussion, Section 4.7. Implications for Shelter Management and Public Policy: o The duplicated sentence of page 20, lines 714-716, regarding APA support (“Future collaboration must focus…”) was removed.
Comments 20: “573-575 yes, though consider also (and consider noting) that owners who microchip their animals may also be more likely to be proactive in locating them if they go missing, thereby increasing RTO among this subset of animals” Response 20: We agree that microchipping likely correlates with broader responsible ownership behaviors, including proactive efforts to locate lost animals. We have added this interpretation to the Discussion to complement the statistical association between microchip presence and RTO. Change implemented: · Discussion, Section 4.7. Implications for Shelter Management and Public Policy (page 21, lines 723-725): o New sentence added: “Microchipping may therefore function both as a technical identification tool and as a proxy for more proactive, responsible ownership, further enhancing the likelihood of RTO for this subgroup.” |
||
|
|
||
Once again, we sincerely thank you for the constructive and insightful suggestions, which have significantly strengthened the conceptual and methodological clarity of our manuscript.
December 17, 2025
